# Correlation of Lower Limb Muscle Activity with Knee Joint Kinematics and Kinetics during Badminton Landing Tasks

**DOI:** 10.3390/ijerph192416587

**Published:** 2022-12-09

**Authors:** Zhe Hu, Youngsuk Kim, Yanan Zhang, Yuxi Zhang, Jiaying Li, Xuan Tang, Jeehoon Sohn, Sukwon Kim

**Affiliations:** 1Department of Physical Education, Jeonbuk National University, Jeonju 54896, Republic of Korea; 2Department of Physical Education, Jeonju University, Jeonju 55069, Republic of Korea

**Keywords:** badminton, single-leg landing, ACL, kinematics, kinetics, EMG

## Abstract

A study on a single-leg landing task after an overhead stroke in badminton suggests that poor knee biomechanical indicators may be a risk factor for anterior cruciate ligament (ACL) injury. A preventive program targeting neuromuscular control strategies is said to alter the biomechanics of the knee joint and have a beneficial effect on reducing ACL injury. However, the relationship between muscle activity around the knee joint and knee biomechanical risk factors in the badminton landing task is unclear. The purpose of this study was to investigate the relationship between this movement pattern of muscle activity and knee kinematics and kinetics. This experiment analyzed knee muscle activity and biomechanical information in a sample of 34 badminton players (17 male, 17 female) during a badminton landing task. We assessed the relationship between the rectus femoris (RF), medial hamstring (MHAM), lateral hamstring (LHAM), medial gastrocnemius (MGAS), lateral gastrocnemius (LGAS), medial and lateral hamstring to quadriceps co-contraction ratio (MH/Q and LH/Q) with the knee flexion angle, valgus angle, extension moment, valgus moment, and proximal tibial anterior shear force. A moderate negative correlation was found between the peak knee flexion angle and electromyography (EMG) activity in LGAS (r = 0.47, *p* = 0.0046, R^2^ = 0.23, 95% CI: 0.16 to 0.70). Peak proximal tibial shear force showed strong and positive correlations with RF EMG activity (r = 0.52, *p* = 0.0016, R^2^ = 0.27, 95% CI: 0.22 to 0.73) and strong and negative correlations with MH/Q (r = 0.50, *p* = 0.0023, R^2^ = 0.25, 95% CI: 0.20 to 0.72). The knee extension moment showed moderate and positive correlations with RF EMG activity (r = 0.48, *p* = 0.0042, R^2^ = 0.23, 95% CI: 0.17 to 0.70) and strong and negative correlations with MH/Q (r = 0.57, *p* = 0.0004, R^2^ = 0.33, 95% CI: 0.29 to 0.76). The peak knee valgus moment showed strong and positive correlations with LH/Q (r = 0.55, *p* = 0.0007, R^2^ = 0.31, 95% CI: 0.26 to 0.75). Our findings suggest that there is a correlation between lower extremity muscle activity and knee kinematics and kinetics during the single-leg landing task in badminton; therefore, lower extremity muscle activity should be considered when developing rehabilitation or injury prevention programs.

## 1. Introduction

Badminton is one of the most popular racket sports. There are more than 160 national associations/federations around the world, attracting many badminton players to participate in the sport. Like other sports, injuries also plague badminton players. Studies have shown that lower extremity injuries account for 40–80% of badminton injuries, and the anterior cruciate ligament is the most serious injury to the lower extremity [1,2]. As we all know, once an injury occurs, it requires a long recovery period, high costs, and unsatisfactory treatment effects, which may lead to serious sequelae, causing many athletes to end their sports careers [3]. The most common injury to the ACL is a non-contact ACL injury (70%), which usually occurs during side cuts, twists, and landings during sports. However, competitive badminton is a complex swing sport involving high-frequency and high-intensity emergency stops, direction changes and landings [4]. Kimura et al. analyzed video and found that landing on one foot after an overhead stroke on the backhand side is the most common action for ACL injuries. This movement requires the badminton player to step back in the direction opposite to the hand holding the badminton racket, take off with the foot on the same side of the hand holding the racket, then land on the foot opposite the hand holding the racket after the overhead stroke at the backhand side of the badminton court [4,5]. Research on single-leg landings after overhead strokes showed that a higher incidence of ACL injury was associated with a higher risk of lower extremity kinematic and kinetic factors during the impact phase, and they concluded that lower knee flexion, higher extension torque, and higher knee valgus angles and moments may be risk factors for ACL injury during single-leg landings after overhead shots [5,6,7]. Because muscle force controlled and regulated by nerve and reflex feedback is the only active regulator of knee load, it is very important to study the relationship between neuromuscular activation and knee biomechanics in the context of ACL injury.

Recent studies have shown that muscle activity in the lower extremity is significantly correlated with knee kinematics and kinetics [8,9]. In a study of a drop task, Brown, T.N. et al. found that pre-activation of the rectus femoris was associated with small peak flexion moments and large proximal tibial anterior shear forces [10]. In addition, in a study on different landing tasks, it was demonstrated that large rectus femoris and gastrocnemius activation was associated with increased ACL loading [11,12,13]. Arampatzis, A. et al., in a study of long-distance runners, showed that reductions in large knee angles were associated with small gastrocnemius activity [14]. In previous studies, the centripetal contraction of the hamstrings produced knee flexion moments and the centrifugal contraction of the quadriceps produced internal extension moments against external moments, and they loaded or unloaded the knee ligaments by modulating their co-contraction [11,15,16]. However, it has been suggested that the co-contraction of the medial and lateral hamstrings affects the ACL differently [17], and in a prospective study it was shown that a large medial hamstring to quadricep co-contraction ratio (MH/Q) was associated with reduced strain by lowering the load on the ACL, and a large lateral hamstring to quadricep co-contraction ratio (LH/Q) was associated with greater ACL load and strain [18]. In addition, in a cross-sectional study, it was shown that greater valgus moments may be associated with greater lateral hamstring activation [19].

In summary, since the association between muscle activity and knee kinematics and dynamics may be different for different tasks, and although there are some recent studies on the kinematics and dynamics of badminton high-risk landing tasks [6,7,20], the research on EMG is still blank, and the relationship between lower extremity muscle activity and knee kinematics and kinetics is unclear. The purpose of this study was to investigate the association between lower limb muscle activity (quadriceps, medial hamstring, lateral hamstring, MH/Q, LH/Q) and knee kinematics (knee flexion angle, valgus angle) and kinetics (proximal tibial anterior shear, extension moment, valgus moment) during a single-leg landing task after an overhead stroke on the backhand side in badminton players. We hypothesized that gastrocnemius muscle activity correlates with the peak extension moment and knee flexion angle. Rectus femoris muscle activity was positively correlated with the knee extension moment and proximal tibial anterior shear force. The medial hamstring to quadricep co-contraction ratio was negatively correlated with the knee extension moment and proximal tibial anterior shear force, and the lateral hamstring to quadricep co-contraction ratio was positively correlated with the knee valgus moment. This study provides a clearer understanding of the relationship between muscle activity and high-risk knee biomechanical parameters, which can provide information to coaches, and athletes when developing training programs to prevent badminton injuries.

## 2. Materials and Methods

### 2.1. Subjects

A total of 34 samples, including 17 males and 17 females, participated in this study. The number of participants was pre-calculated from the experimental work using G* Power 3 software to provide α = 0.05, 80% statistical power and an effect size of 0.4.

All participants were recruited by Jeon Buk University, with specific inclusion criteria as follows: (1) An experienced physiotherapist to determine by observation and brief assessment that there is no significant restriction of movement or muscle weakness. (2) No lower extremity pain before the test. (3) Subjects are required to participate in organized training at least four times a week. The study was approved by the Ethics Committee of Jeon Buk University (JBNU2022-01-004-002). Before participating in this study, all subjects were informed about the trial procedure, and read and signed an informed consent form.

### 2.2. Prepare for Testing

Trial data collection Thirteen infrared cameras (Prime 17W, OptiTrack, Natural Point, Inc., Corvallis, OR, USA) were used to capture the kinematic data of each participant at a sampling rate of 120 Hz. In the experiment, the matching marker was a 14 mm reflective marker, and each subject was marked with 28 reflective skin markers, including 18 bony markers, 6 calibration markers, and 4 markers that distinguish between left and right thigh and shin segments [21,22]. Ground reaction force data was collected at 1200 Hz using an OR6-6-2000 force platform (AMTI Inc.) from Newton, MD, MA, USA, with a maximum delay time of 6 ms. The EMG collection system (Trigno Avanti Sensor, Delsys, Natick, MA, USA) was selected as the EMG data acquisition device. For the EMG signal acquisition, we used a Trigno Avanti sensor (Delsys, Natick, MA, USA; 3.7 cm × 2.7 cm). All EMG sensors (Trigno Avanti Sensor) had a common-mode rejection ratio of 80 dB and were synchronized with kinematic and kinetic data by Motive (OptiTrack, Natural Point, Inc., Corvallis, OR, USA), with EMG acquisition frequency of 1200 Hz. Surface electrodes were selected from the quadriceps femoris (rectus femoris), medial hamstrings (semitendinosus), lateral hamstrings (biceps femoris), medial gastrocnemius, and lateral gastrocnemius. All EMG reference standards were based on Marco Barbero et al. [23].

The location selection is as follows: 40% (RF) of the line connecting the superior edge of the patella to the anterior superior iliac spine. 80% of the line connecting the medial popliteal crease to the ischial tuberosity (MHAM). 80% of the line connecting the lateral popliteal ridge to the ischial tuberosity (LHAM). 85% of the line connecting the medial Achilles tendon to the medial hamstring (MGAS). 85% of the line connecting the lateral Achilles tendon to the lateral popliteal crease (LGAS). The hair on the skin surface was shaved and cleaned with alcohol before the electrodes were attached. After the skin was dry, the EMG electrodes were attached. At the same time, motion tape was used to fix the electrodes and reduce motion artifacts [24]. The maximum voluntary isometric contraction (MVC) test was performed on each muscle for 5 s in the following manner: prone position, knee flexion 45 degrees, extension band fixed on the back of the ankle, knee flexion action (HAM) [9]. Sit in a sitting position, bend your knees 90 degrees, fix the extension strap on the front of your ankle, and perform a knee extension (QUA) [9]. Lie prone, knees straight, stretch straps fixed on the forefoot, and perform plantar flexion (GAS) [25]. Use Fengcai’s badminton server SPT6000 (SPTLOOKER, Guangzhou, China) to send the shuttlecock to the designated area in the same state. Subjects wear uniform shorts, individual socks and shoes, and use uniform rackets.

### 2.3. Test Procedure

The laboratory design is as shown in Figure 1. The badminton serve position (A) is located at the intersection of the center line and the serve line, and the badminton is sent to the designated area B (50 cm × 50 cm) in the same state through the badminton serve machine.

Subjects performed a 10 min warm-up (jogging, swinging), and then performed the single-leg landing experiment task of badminton. From the starting position, the subject simulates a step back towards the left rear backhand side of the court, and after performing an overhead stroke, the left leg lands on the force plate, and they quickly return to the starting position. Subjects simply hit the shuttlecock in their customary manner to the opposite back-side court area C (220 cm × 80 cm). Subjects were allowed to perform multiple exercises followed by three to five consecutive trials. The main movements are shown schematically in Figure 2. To avoid any instructive influence on the natural performance of the subjects, no other instructions related to stepping, landing or stroke technique were provided.

### 2.4. Data Processing and Analysis

GraphPad PRISM 8.0 (GraphPad Inc., San Diego, CA, USA) was used to correlate lower extremity muscle activity with biomechanical variables of the knee joint that are thought to influence the risk of ACL injury. Lower extremity muscle activity variables included mean values of RF, MHAM, LHAM, MGAS, LGAS muscle activity and co-contraction ratios of MH/Q and LH/Q. Biomechanical variables included peak knee flexion, valgus angle, extension and valgus moments, and proximal tibial anterior shear forces during the impact phase (i.e., within 100 mms after initial contact, as ACL injuries occur within this time frame). Correlations were performed between the mean values of RF, HAM, and GAS muscle activity and the peak knee flexion angle and extension moments and proximal tibial anterior shear forces, as well as between the co-contraction ratios of MH/Q and LH/Q and the peak extension and valgus moments and proximal tibial anterior shear forces. The 95% confidence interval (CI) was used for quantitative description. For the degree of correlation, the Pearson correlation coefficient (r) was used for the parametric test, and the Spearman rank correlation (r) was used for the nonparametric test. In addition, the coefficient of determination (R2) is used in parametric data to represent the amount of variation in a screening test. As described by Llurda et al., correlation strengths were classified as (0–0.3) small, (0.3–0.5) moderate, (0.5–0.7) strong and (0.7–1) very strong [9].

## 3. Results

The mean and standard deviation of subject information are as follows: females were 20.80 (±2.12) years old, with a height of 1.67 (±0.05) m, and a mass of 59.45 (±5.84) kg, and males were 20.70 (±1.42) years old, with a height of 1.78 (±0.06) m, and a mass of 71.82 (±9.31) kg.

The mean and standard deviation of the EMG parameters during the landing phase of the single-leg landing task after a backhand-side overhead stroke in badminton are shown in Table 1.

The average peak flexion angle of the knee joint during the landing impact phase was 46.42 ± 9.95°, and the peak valgus angle was 7.82 ± 6.21°. The mean peak extension and valgus moments were 0.89 ± 0.46 Nm/kg/m and 0.22 ± 0.17 Nm/kg/m. The mean peak normalized anterior tibial shear force was 2.02 ± 0.89 N/kg (Figure 3).

During the impact phase of the badminton landing task, peak knee flexion angle was found to be moderately negatively correlated with LGAS EMG activity (r = 0.47, *p* = 0.0046, R^2^ = 0.23, 95% CI: 0.16 to 0.70). Peak proximal tibial shear was strongly positively correlated with RF EMG activity (r = 0.52, *p* = 0.0016, R^2^ = 0.27, 95% CI: 0.22 to 0.73) and strongly negatively correlated with MH/Q (r = 0.50, *p* = 0.0023, R^2^ = 0.25, 95% CI: 0.20 to 0.72). Knee extension moment was moderately positively correlated with RF EMG activity (r = 0.48, *p* =0.0042, R^2^ = 0.23, 95% CI: 0.17 to 0.70) and strongly negatively correlated with MH/Q (r = 0.57, *p* = 0.0004, R^2^ = 0.33, 95% CI: 0.29 to 0.76). Peak knee valgus moment was strongly positively correlated with LH/Q (r = 0.55, *p* = 0.0007, R^2^ = 0.31, 95% CI: 0.26 to 0.75). Please refer to Figure 4.

## 4. Discussion

The results of this study identified a significant correlation between muscle activity and knee kinematics and kinetics during a high-risk landing task in badminton. This is consistent with our hypothesis that our main findings are: 1. Lateral gastrocnemius muscle activity was negatively correlated with peak knee flexion angle. 2. Peak extension moment was positively correlated with rectus femoris muscle activity and negatively correlated with MH/Q. 3. Peak proximal tibial shear force was positively correlated with rectus femoris muscle activity and negatively correlated with MH/Q ratio. 4. Peak knee valgus moment was positively correlated with LH/Q. 

During the landing task, a small knee flexion angle is considered a risk factor for ACL injury [26]. The analysis of videos during non-contact ACL injuries showed that the injured individuals exhibited a more extended knee landing position (i.e., smaller knee flexion angle). In studies targeting anatomical structures, smaller knee-flexion angles were associated with larger ACL elevation angles, and larger ACL elevation angles would result in greater strain on the ACL [27]. In studies of different landing, jumping, and lateral cutting tasks, smaller knee angles exhibited greater ground reaction forces and greater quadricep loading forces compared to larger knee angles [28]. Additional research has shown that anterior tibial shear forces generally occur at smaller knee angles, and this has further been established in our study, and a recent study on a single-leg landing task showed that smaller knee angles are predicted to be associated with an increase in ACL loading [29]. Therefore, it is generally accepted that either increasing the initial contact knee flexion angle or the peak knee flexion angle during landing is safer for the ACL. Our findings suggest that lateral gastrocnemius muscle activity is negatively correlated with peak knee flexion angle. Large lateral gastrocnemius activation may increase the risk of ACL injury, which may be because the gastrocnemius is a bi-articular muscle that spans both the knee and ankle joints, attaching proximally to the posterior femur and ending distally at the heel tuberosity, which increases gastrocnemius activation during the landing phase to absorb the impact of the landing phase and maintain ankle joint stability. Studies have shown that at smaller knee angles, large gastrocnemius activation may increase muscle stiffness to produce a posterior force on the distal femur, resulting in the extension of the knee joint (reduced flexion angle) [30]. This is consistent with another finding of ours regarding the correlation of the gastrocnemius muscle. M. Sharifi, through a study of finite-element skeletal models of the lower extremity, concluded that a small gastrocnemius force and a large knee angle are decisive parameters for knee stability [31]. Other studies have shown that increased gastrocnemius muscle activation at smaller knee angles will increase ACL stress [32]. Therefore, reducing gastrocnemius muscle activity may be helpful in preventing ACL injury.

Previous studies have demonstrated that increased anterior shear forces are associated with increased loading of the ACL [33]. This is primarily explained by the fact that the ACL originates from the medial aspect of the femoral lateral condyle and ends at the anterior aspect of the tibial intercondylar eminence, and its primary function is to maintain knee-joint stability by providing a horizontal posterior component to prevent forward movement of the tibia [34]. From the mechanical point of view, if a pulling force is generated at the ACL that exceeds the ultimate loading capacity, then the ACL will become damaged. Our study found a positive correlation between quadricep and proximal tibial anterior shear force. This implies that increased muscle activity of the quadriceps may be associated with increased loading of the ACL, which is consistent with the findings of Withrow, T.J. and Navacchia, A. et al. in studies on cadavers and for musculoskeletal models [11,35], and in studies on functional tasks by Brown, T.N. and Maniar, N. et al., who also demonstrated a positive correlation between rectus femoris activation and proximal tibial anterior shear force [10,36]. Since the end of the distal quadriceps is connected to the anterior aspect of the proximal tibia through the patellar ligament, the patellar tendon tibial axis angle [37] and the ACL elevation angle (the angle between the ACL and the tibial plateau) [38] increase simultaneously at smaller knee angles, which implies an increase in the horizontal component of the knee extension force generated by quadricep contraction, i.e., an increase in the proximal tibial anterior shear force. Therefore, the greater the quadricep activation associated with greater anterior shear forces, the higher the loading on the ACL to counteract the anterior shear forces, and potentially the greater the risk of ACL injury. In contrast, Morgan, K.D. et al., in their analysis of a study of single-leg jump landing tasks in soccer players, suggested that quadricep activation during landing may represent a generalized muscular strategy to increase knee stiffness and protect the knee and ACL from external knee loading and risk of injury [39]. However, this study also needs to consider the role played by the hamstrings in this, and increased quadricep activation cannot be considered as a strategy to protect the ACL.

The results of another study showed that the quadricep (rectus femoris) muscle was positively correlated with large extension moments, which is similar to the findings of Brown, T.N. et al. in the landing task [10]. It is well known that the function of neuromuscular contraction is to maintain body balance and secondarily to maintain stability. When badminton players land during the early stages of the task, the impact force is large, and to maintain body balance during the landing process and control the rate at which the body’s center of gravity falls, it is then necessary for internal muscle contraction to produce sufficient extension moments to balance the flexion moments produced by external forces. The quadriceps have been shown to be the main contributor to the extension moment [40]. Poor neuromuscular control patterns in the maintenance of balance may be overly dependent on the greater extension moment generated by the contraction produced by quadricep activation, and the greater extension moment is one of the determinants of greater anterior knee forces that may lead to increased ACL loading [12]. However, we need to note that although a greater extension moment is one of the determinants of increased tibial anterior force, this does not mean that large quadricep activation determines tibial anterior shear force, as it is also influenced by other factors (e.g., angle, etc.) [30]. Therefore, our results can only suggest that large quadricep activation with a large extension moment may be associated with the risk of ACL injury.

The results of one of our major studies suggest that large MH/Q co-contraction ratios are associated with small extension moments and small proximal tibial anterior shear forces. Heinert, B.L., in a study examining a single-leg drop jump task, determined that within 100 ms after toe touch, the low-ratio group showed an increase of 16.6% in ligament shear force, a 26% increase in tibiofemoral shear force, and a 6% increase in vertical force between the femur and tibial plateau compared to the high H/Q ratio group [41]. Rachel Dedinsky, in a review article, argued that if sufficient H/Q ratios are produced, the ACL strain stress is reduced and the risk of ACL injury is decreased [42].

Meniar, N. and Toor, A.S. et al. demonstrated the importance of the medial hamstrings in knee stabilization [33,43], Maniar, N. demonstrated in a study of cadavers that the medial hamstrings and flounder were the muscles that produced the greatest posterior shear forces against anterior shear force [33].

Toor, A.S., in another study of cadavers, obtained results which showed that the medial hamstrings are involved in rotation, translation and internal/external rotation control of the knee joint and that applying anterior, external rotation and external rotation forces to a hamstring-deficient knee significantly increased motion in these planes [43]. In addition, Benjamin G Serpell et al. demonstrated under computed tomography (CT) that increasing the MH/Q co-contraction ratio was associated with reduced strain in the ACL [18], so we can assume that increasing the MH/Q co-contraction ratio may be beneficial in preventing the occurrence of ACL injuries. Our other result suggests a positive correlation between large LH/Q and large knee valgus moment, and some studies have demonstrated that increasing the valgus moment increases ACL loading [44]. This is similar to studies by Annemie et al. [45] and Serpell, B.G. et al. [18]. In a prospective study, Annemie et al. showed that athletes already exhibited stronger neuromuscular activation patterns in the lateral hamstrings prior to ACL injury [45]. The results of Serpell, B.G. et al. demonstrated that lateral hamstring–quadricep co-activation was related to ACL elongation [18]. Therefore, reducing the LH/Q ratio may increase the risk of ACL injury.

Our article has several limitations. Both EMG and 3D motion analysis data acquisition processes have limitations. The EMG signal is highly complex, stochastic, and susceptible to the inherent characteristics of the individual. Model measurement errors, including misaligned knee markers, errors in skin motion artifacts, definition of joint center position, and leg length discrepancies may affect kinematic results [46,47,48]. Although we found a correlation between lower-limb muscle activity and knee-joint kinematic and kinetic variables, it does not imply a cause-and-effect relationship. It is important to note that our study design does not allow for causal inference. This finding should not be interpreted to mean that lower-limb muscle activity causes changes in kinematics and/or kinetics in single-leg landings in badminton. The badminton landing task is performed with the support of a kinetic chain between the arm, trunk, pelvis, thigh, shank and foot segments, either up to down or down to up, which may alter the biomechanical characteristics of the knee joint. In this study, we only considered the relationship between the muscles surrounding the knee joint and the knee biomechanical factors during the impact phase; however, deficiencies in the neuromuscular control of each segment may lead to suboptimal movement patterns and thus increase the risk of ACL injury [49,50,51]. Therefore, we suggest that further studies should include muscles that do not cross the knee joint, such as the trunk muscles, gluteus maximus, gluteus medius and soleus muscles, to influence knee kinematics and kinetics to expand our understanding of the relevance of neuromuscular activation patterns of more muscles to knee-joint dynamic control. It is also worth noting that although we found a correlation between RF muscle activity and proximal tibial forward shear, our study found that forward shear generally occurs at a very early stage of landing (Figure 3e), which may be too late to respond through reflexive muscle activity, and therefore reducing muscle activity in RF during the impact phase may be less appropriate as a strategy for developing training. This relationship between muscle activity and proximal tibial anterior shear force is more likely to be influenced by pre-landing preparation, so the next studies may need to consider more the effect of muscle activity patterns on tibial anterior shear force in badminton before landing, which would be more beneficial in developing injury prevention strategies.

Despite these limitations, to our knowledge, this study is the first to investigate the correlation between lower-limb muscle activity and knee biomechanics during badminton landing tasks. Our findings suggest that the biomechanical characteristics of the movements required for individual sports may differ, the factors contributing to injury may differ, and the information required to provide injury prevention strategies may differ. This is similar to what was reported in the studies of Alcantarilla-Pedrosa, M. et al. [52] and Demeco, A. et al. [53]. Alcantarilla-Pedrosa, M. et al. [52] showed in a video analysis of two seasons of a Spanish professional soccer team that due to the different positions (fullbacks, central-midfielders, central backs, wide-midfielders), the required motor characteristics differed, leading to different risk and severity of injury. In a review study of padel, Demeco, A. et al. also reported common injuries to their lower and upper extremities related to their change of direction movements and overhead strikes during the sport [53]. Through the above studies, we suggest that before developing injury prevention strategies, it can be useful to understand the sport characteristics and injury risk of different sports or different athletes of the same sport (e.g., soccer) through video analysis of the game, and through clinical and instrumental (e.g., surface electromyography and inertial measurement unit) assessment of athletes with possible deficits in specific movements (motor and neuromuscular control deficits). The identification of biomechanical characteristics of individual sports in the required movements targeted to develop injury prevention strategies may play an important role in the management of athletes.

## 5. Conclusions

Our findings suggest that there is a correlation between lower-extremity muscle activity and knee kinematics and kinetics during the single-leg landing task in badminton; therefore, lower-extremity muscle activity should be considered when developing rehabilitation or injury prevention programs.

## Figures and Tables

**Figure 1 ijerph-19-16587-f001:**
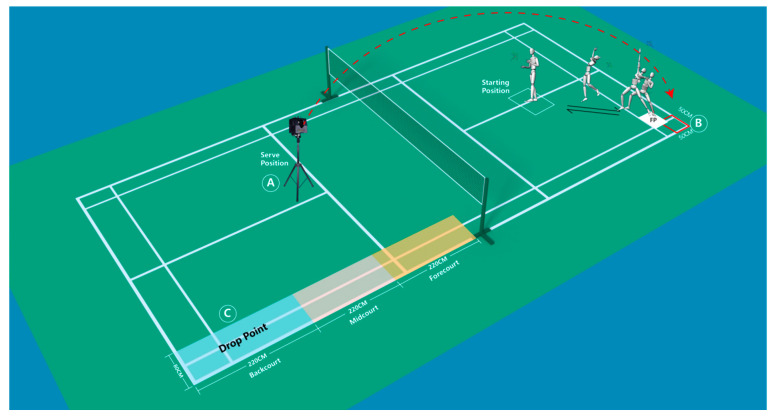
Experimental setup. Force plate (FP) and badminton serve machine position. The badminton serve machine sends shuttlecocks from area A to area B which is 50 × 50 cm. The subject steps back from the starting point in a left–back direction, then jumps and performs an overhead strike. The subject performs a single-leg landing on the force plate and then returns to the starting position. C area is the ball drop point after hitting the ball.

**Figure 2 ijerph-19-16587-f002:**
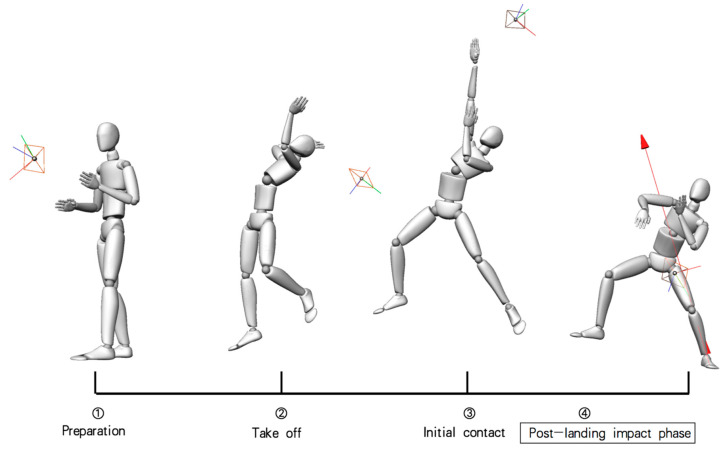
Events of badminton single-leg landing task.

**Figure 3 ijerph-19-16587-f003:**
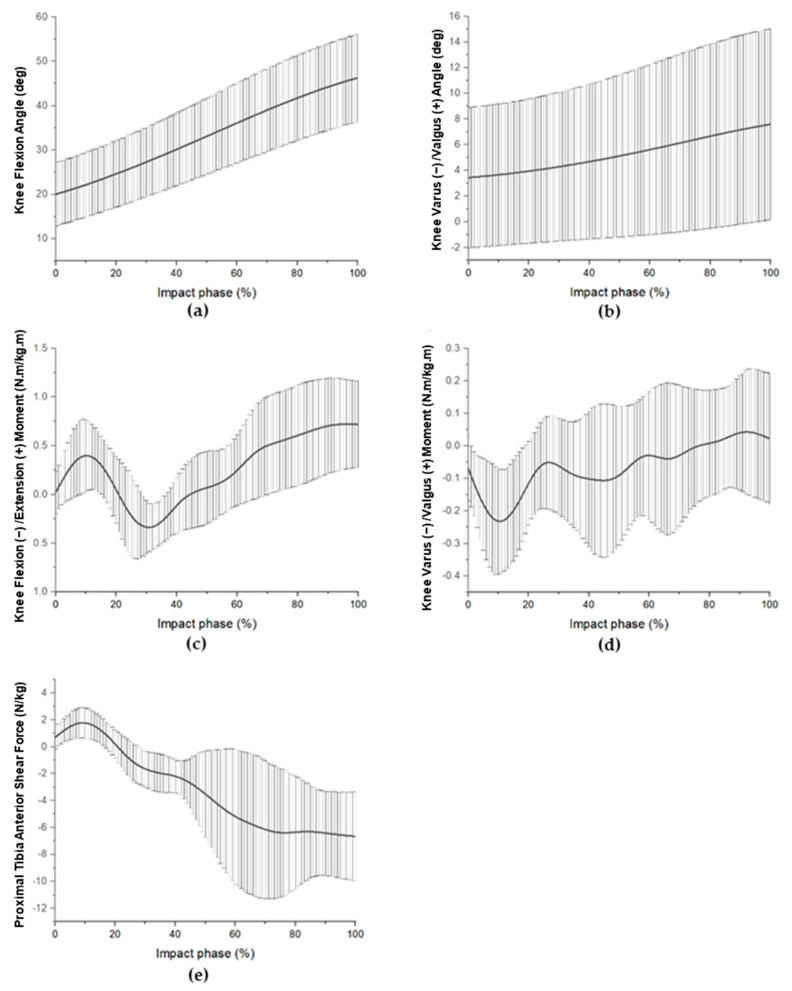
Mean (±SD) knee biomechanical patterns during the impact phase of the badminton landing task. The flexion angle (**a**) and valgus angle (**b**), extension moment (**c**) and valgus moment (**d**), and anterior proximal tibia shear force (**e**) are shown.

**Figure 4 ijerph-19-16587-f004:**
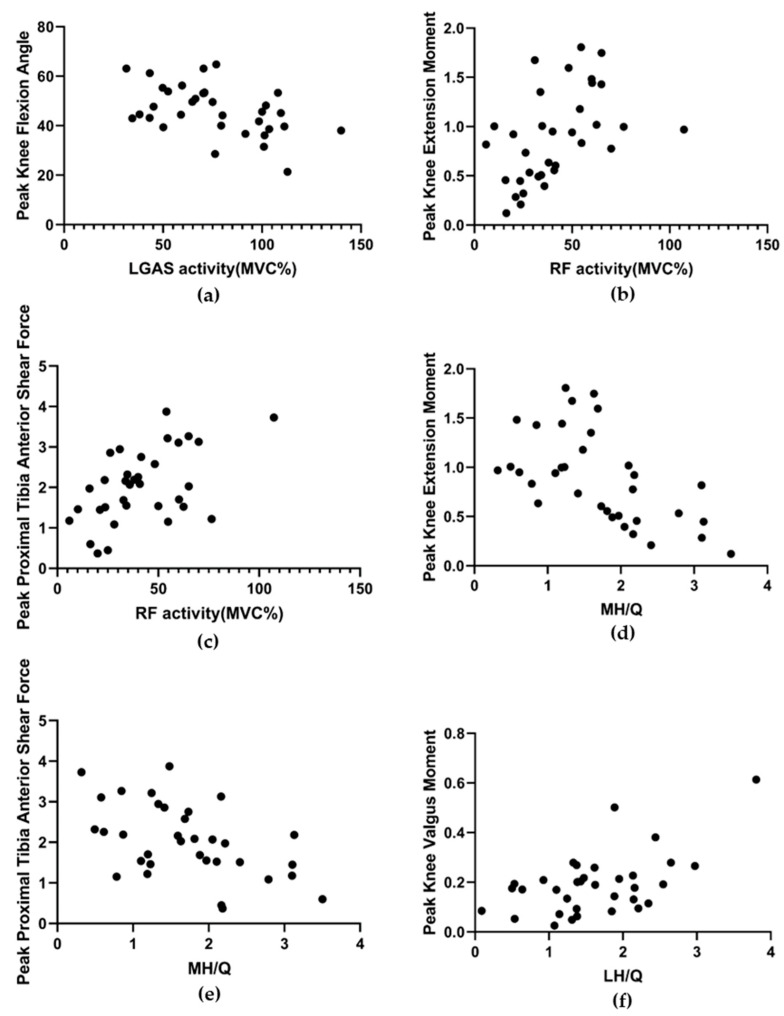
Relationship between LGAS activity and peak knee flexion angle (**a**). Relationship between RF activity and peak extension moment (**b**) and peak proximal tibial anterior shear force (**c**). Relationship between MH/Q and peak extension moment (**d**) and peak proximal tibial anterior shear force (**e**). Relationship between LH/Q and peak valgus moment (**f**).

**Table 1 ijerph-19-16587-t001:** Average knee muscle activity during the impact phase of badminton landing task.

Variable	Activity (% of MVIC or Ratio)
Medial hamstrings (MHAM)	40.18 ± 17.66
Lateral hamstrings (LHAM)	17.89 ± 9.38
Medial gastrocnemius (MGAS)	86.65 ± 41.10
Lateral gastrocnemius (LGAS)	81.52 ± 36.54
Rectus Femoris (RF)	40.95 ± 25.80
MH/Q	1.70 ± 0.80
LH/Q	1.62 ± 0.78

## Data Availability

The data presented in this study are available on request from the corresponding author.

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
