# Peer review of "Correlation of Lower Limb Muscle Activity with Knee Joint Kinematics and Kinetics during Badminton Landing Tasks"

_ijerph, 2022, doi:10.3390/ijerph192416587_

Round 1

Reviewer 1 Report

It's an interesting study addressing an important and relevant question. The manuscript is well written and easy to follow and understand. However, there are some issues that needs to be addressed.

1. The major concern is the data treatment and processing. It seems like the authors have made multiple analysis using Pearson correlation coefficient on multiple variables. Such an approach implies that the p-value must be adjusted (reduced) to avoid false correlations. Multiple regression analysis would have been more appropriate.

2. Minor concern. Under section 2.1 there is some unclear information regarding the descriptive data presentation. Provide an explanation of what the subjects numbers mean. Are they mean +/- standard deviation or standard error or are they median +/- standard deviation or  standard error?

3. Between the lines 165 - 179 there seems to be remains of the instructions to the authors. Should be removed.

Author Response

Thank you for your thorough review. Please check the file for the detailed revisions.

Reviewer 2 Report

The introductuon, methodology, and conclusions section needs more elaboration. However, I am satisfied with the paper, and agree with its publication.

Author Response

Thank you for your thorough review. Please check the file for the detailed revisions

Reviewer 3 Report

Dear Authors,

In my opinion, the topic is interesting considering the emerging role of biomechanics in the physical training field. However, I have several concerns regarding the methodological implant of this study, and several critical issues should be addressed.

Major revisions:

MATERIALS AND METHODS: The study design should be clarified.

MATERIALS AND METHODS: The number of patients included in the study and related baseline characteristics should be reported in the Results section. Please move this part to the suggested section.

MATERIALS AND METHODS: In the Eligibility criteria, “unrestricted range of motion in any joint” was mentioned. Was this determined following a range of motion assessment by qualified personnel or is it based only on observation? Please, clarify this point in the inclusion criteria.

MATERIALS AND METHODS: Outcomes considered in this study should be reported. I suggest providing a specific subsection.

MATERIALS AND METHODS: The Authors should report who performed the analysis and how conducted the trials (qualification, degree).

RESULTS: The number of patients included in the study and related baseline characteristics should be reported in this section.

DISCUSSION: It could be useful to insert information about possible prevention strategies based on the findings of this study. For example, the prevention of injuries by studying better the biomechanics of the movements required by individual sports could play a key role in the athlete’s management. Another important topic is the specific demands of the sport, which can be evaluated through match analysis.

According to this, you should cite the following references:

-        Alcantarilla-Pedrosa M, Álvarez-Santana D, Hernández-Sánchez S, Yañez-Álvarez A, Albornoz-Cabello M. Assessment of External Load during Matches in Two Consecutive Seasons Using the Mediacoach® Video Analysis System in a Spanish Professional Soccer Team: Implications for Injury Prevention. Int J Environ Res Public Health. 2021; 18(3):1128. doi: 10.3390/ijerph18031128.

-        Demeco A, de Sire A, Marotta N, Spanò R, Lippi L, Palumbo A, Iona T, Gramigna V, Palermi S, Leigheb M, Invernizzi M, Ammendolia A. Match Analysis, Physical Training, Risk of Injury and Rehabilitation in Padel: Overview of the Literature. Int J Environ Res Public Health. 2022;19(7):4153. doi:10.3390/ijerph19074153.

-        Xu D, Cen X, Wang M, Rong M, István B, Baker JS, Gu Y. Temporal Kinematic Differences between Forward and Backward Jump-Landing. Int J Environ Res Public Health. 2020; 17(18):6669. doi: 10.3390/ijerph17186669.

Minor revisions:

MATERIALS AND METHODS: Line 165-179. This section still contains the Instruction for Authors included in the IJERPH template file. Please, remove this part.

MATERIALS AND METHODS: Figure 1 is difficult to read. Please, improve this Figure.

Author Response

(The authors gave the same response as above.)

Round 2

Reviewer 1 Report

I feel that the manuscript has improved and can be accepted for publication

Reviewer 3 Report

Dear Authors,

In my opinion, the topic is interesting and actual, and the manuscript is well written.

The results are intriguing and might significantly improve knowledge in this field.

You have significantly improved the manuscript following my suggestions, and for this reason, in my opinion, the article is suitable for the Journal.